# A semi-empirical model for near sea surface wind speed deficits downstream of offshore wind parks fitted to satellite synthetic aperture radar measurements

Johannes Schulz-Stellenfleth and Bughsin Djath Helmholtz-Zentrum Hereon, Max-Planck-Str. 1, 21502 Geesthacht, Germany **Correspondence:** Johannes Schulz-Stellenfleth (johannes.schulz-stellenfleth@hereon.de)

**Abstract.** A two-dimensional advection/diffusion model for the near sea surface wind speed deficit downstream of offshore windparks is fitted to satellite synthetic aperture radar (SAR) data. The Wake2Sea model enables the inclusion of offshore wind farm (OWF) wake effects in existing atmospheric model data at low computational costs and employs the standard Fitch parameterisation to describe the momentum sink associated with wind turbines. Model wind fields from the German weather

- 5 centre are used as prior information about the unperturbed atmosphere without OWFs. Using 30 Sentinel-1A/B satellites SAR scenes acquired over the German Bight representing different stability and wind speed regimes, a 4DVAR scheme is applied to optimize the agreement between simulated and observed radar cross sections. The method adjusts 8 parameters in the wake model and also applies corrections to the background wind field on a spatial scale of 40 km. An L-curve analysis is applied to choose the weighting of prior knowledge and observations in the cost function. The method improves the match
- 10 between observations and simulations significantly, if uncorrected model wind fields are used as a baseline. Furthermore, the inclusion of the empirical wake model leads to improvements when the background corrected wind field is used as a reference. Comparisons with data measured at the fixed platform FINO-1 adjacent to the first German offshore wind park Alpha Ventus, showed that the proposed inclusion of wakes in the atmospheric model data leads to a significantly improved match.

# 1 Introduction

- 15 The global installed offshore wind energy power has increased about tenfold over the last decade reaching 64 GW in 2023 (WFO, 2023). With a share of about 47% China is currently the largest offshore wind farm (OWF) operator worldwide. Some studies suggest that over 380 GW of new offshore wind capacity will be added over the next decade globally (Williams and Zhao, 2023). In Europe the United Kingdom is the country with most installations followed by Germany with 14 GW and 8 GW installed power by 2023 respectively. According to the European Union (EU) Strategy on Offshore Renewable Energy,
- 20 the installed offshore wind power in Europe will grow from about 28 GW in 2022 to about 60 GW by 2030 (EU, 2021). In Germany the goal to achieve 70 GW offshore wind energy by 2045 is written in law (Deutscher Bundestag, 2024).

It is obvious that these rapid developments come with a large spectrum of challenges in the economic, political and research sector. A large number of studies already exists, which analyze the impact of offshore wind farms on the atmosphere, often with a focus on wakes in the atmospheric boundary layer (ABL) (Siedersleben et al., 2020; Akhtar et al., 2021; von Brandis et al.,

- 2023; Platis et al., 2018). One reason for this interest is the direct implications of these wakes for the optimisation of power 25 yields considering shadowing effects, as well as the role of turbulent wakes for the fatigue loading on downstream turbines. The respective processes in the ABL have been studied with different types of numerical models including mesoscale models (Siedersleben et al., 2020), Large Eddy Simulation (LES) models (Vollmer et al., 2017) and industry models (Cañadillas et al., 2020). Furthermore, different types of observation techniques were applied, e.g. Light Detection and Ranging (LIDAR) systems
- (Schneemann et al., 2020) and spaceborne synthetic aperture radar (SAR) sensors (Djath and Schulz-Stellenfleth, 2019). The 30 existing studies show that OWF wakes can extend well above 100 km downstream in cases where the ABL is very stable. Typical wind speed deficits are in the range 10%-20% (Djath et al., 2018). There is ongoing research about atmospheric wakes, e.g. concerning the interaction of wakes, or the coupling with coastal effects (Djath et al., 2022; Schulz-Stellenfleth et al., 2022). Furthermore, there is still debate about optimal parameterisations of OWFs in numerical models (Fischereit et al., 2022;
- 35 Ali et al., 2023).

In addition to the OWF effects in the ABL, potential impacts in the ocean have gained growing attention (Christensen et al., 2013; Broström, 2008; Christiansen et al., 2022; Daewel et al., 2022). There are basically two types of processes discussed in literature so far:

- Effects caused by the modified wind forcing at the sea surface (Christiansen et al., 2022; Daewel et al., 2022).
- Effects related to the interaction of the water with the OWF foundation structures (Christensen et al., 2013; Grashorn 40 and Stanev, 2016; Carpenter et al., 2016; Carpenter and Guha, 2024).

The present study is connected to the modelling of the first type of processes, where accurate estimates of near surface wind speeds in the surroundings of OWFs are required. As mentioned above, most studies concerned with OWF wakes in the ABL have a focus on the impacts around the hub height, which are most relevant for power yields. Near surface wind speeds around 45 OWFs modelled with mesoscale models have been used to drive ocean models (Christensen et al., 2013), but very little has been done concerning the validation of these data. On the other hand, there is a large amount of satellite SAR data available, which provide two-dimensional (2D) information on near ocean surface wind speeds with high spatial resolution (Lehner et al., 1998), but the condensation of respective information on OWF wakes in parameterised form is still at a very basic level (Djath et al., 2018). Against this backdrop, the main objectives of the present study are as follows:

- 50 - Condense the OWF wake information contained in SAR data in a 2D semi-empirical model, which captures the main characteristics, but has small computational demands compared to a 3D atmospheric model
  - Design this model as a tool for ocean modellers, to generate wind forcing for OWF impact studies, allowing the consideration of a multitude of OWF scenarios not feasible with 3D atmospheric models.

The proposed semi-empirical model can be used to add OWF wakes to existing atmospheric model data sets. Many of these data sets, like ERA5 (Hersbach et al., 2020), are intensely used as references by the scientific community and the proposed 55 tool can massively enhance the applicability of these data in the OWF context.