# Peer review of "A semi-empirical model for near sea surface wind speed deficits downstream of offshore wind parks fitted to satellite synthetic aperture radar measurements"

_Wind Energy Science, 2025_

## Referee Comment (RC1)

The work described in "A semi-empirical model for near surface wind speed deficits downstream of offshore wind parks fitted to satellite synthetic aperture radar measurements" is of great interest as it relies on large-scale satellite observations to improve the accuracy of wake modelling in datasets from numerical atmospheric models. SAR imagery is indeed highly valuable information about the surface wind field in coastal and offshore locations and deserves to be better known in the offshore wind community.

While the scientific aspect and the innovative source of data are of great interest to the research community, the paper lacks consistency in the final application and the description of the methodology. Indeed, the authors emphasize that the final application is related to the improvement of surface wind speeds for the forcing of ocean models. However, the context of ocean modelling is rather poorly described, the focus being made by the authors on the wind speed in altitude and offshore wind energy in general.

Secondly, the description of the global methodology needs to be reviewed with particular care on the uniformisation of notations, precise mention of the considered height at each step of the process. Mixing parameterisations and concepts at different altitudes makes it complicated to follow the methodology and makes the whole process questionable with different extrapolations concepts in waked flows that are not properly discussed. The authors try to implement a rather simplistic approach based on some parameters to account for the change in altitude into their model. Indeed, the use of Fitch parameterisation and Ct requires a wind speed in altitude, the input of their model is a 10-m wind speed which is scaled up to address a wind field in altitude. But the output of the model is a surface wind field with inclusion of a wake. The model is trained/optimised on the difference between backscattered signals which are supposed to represent the surface wind field. A lot of uncertainties due to the change in altitude embarked in the whole methodology makes the conclusions of the author (i.e. an absence of turbulence modelling in their model to extrapolate properly insitu data which measurement heights are higher than 10m) hard to validate.

Thirdly, the authors use satellite imagery. The description of this remote sensing technique, as well as the resulting data, is very interesting and brings a new horizon to the offshore wind community. However, the offshore wind community is not familiar with this data, and it is important to demonstrate that the SAR-derived wind field does not bring additional uncertainties. I advise the authors to properly describe their methodology with the SNAP tool distributed by ESA and to explain why they do not use the L2 OCN products that include wind speed, NRCS and quality flags, that are provided and validated by the CALVAL of the Sentinel-1's Mission Performance Center.

I therefore recommend a **major revision** of this paper.

Detailed comments on the document:

A general comment on the position of the figures in the article: it will help to have the figures closer to the paragraph mentioning and explaining the figures.

**Section 1: Introduction**

Line 28: Please replace "industry" with "engineering".

Line 54: You could emphasize that the addition of the wakes is done a-posteriori to existing surface datasets.

Line 56: The application of the tool is in contradiction with Lines 64-65. This sentence needs to be discussed and rephrased. The OWF context generally means wind speed or yield assessment in altitude, or structural analysis (from floaters/foundations up to the upper tip of the blades).

Figure 1: In general, color bars may be easier to understand with the title next to it.

- Figure 1c: Ct curve: wind speed at hub height or rotor-averaged wind speed. Is Ct extended for $u\_hub > 25m/s$ with a stall to a zero-Ct value?
- Figure 1d: I am not sure to understand the relevance of this graph. Would it be more interesting to locate the power of the turbines on the map as in a and b? Please describe or introduce dZ notation.

Line 59: Discuss the choice of the SAR scenes and how only 30 is relevant to cover all situations.

Line 61: The reference to the Fitch parameterisation needs to be added.

Line 61: Please uniformize the notation of the thrust coefficient, Ct or CT, throughout the paper.

Line 66: Explicit the height of the "near surface wind". Is it 10m?

**Section 2: Wake2Sea**

In general, mathematical formulations need to be uniformized throughout the paper. Notations and parameters need to be introduced and clearly defined at each step.

When dealing with Ct and the wind speed U, it is not always clear at which height the equations are considered (10m or hub height). The applicability of Fitch parameterisation at 10m with Ct defined with $U\_10m$ needs to be strongly discussed.

For clarity, it would help to:

- Mention at the beginning of Section 2 that the eight constant parameters to be fitted will be called \alpha_i€(1-8) in the following equations;
- Recall the final system of equations in Line 130.

Line 74: Rephrase "bottom layer of the atm above the ocean surface".

Line 75: Is U at 10m? Within the surface layer? Within the whole MABL?

Line 80: Define $\Chi = \nu_v/dZ^2$ as the vertical diffusion parameter.

Line 81: Uniformize upper or lower case for u+/u-, and U in Lines 75 and 81. Line 82: Please recall the Fitch parameterisation (its mathematical formulation) as it would make it easier to understand the following equations.

Lines 94 and 95: Please discuss the simplifications with order of magnitude.

Line 109: Define $\Delta T$: is it $T\_2m$ – SST?

Line 110: Introduce \alpha.

Line 121: Ct(U) with U at hub height. I am not sure that Ct(U downscaled to 10m) means anything.

Line 126: Please discuss this assumption, especially the applicability of the Fitch parametrization at 10m.

Line 136: dZ is related to the height of the MABL? Is it constant for all stability conditions? A recall of the standard Fitch parametrization would be easier to understand if recalled in Line 82 to introduce the concepts and notations, especially of dZ and the "bottom layer" which seems to be a modelling concept rather than a physical part of the MABL.

Table 1: The table could be placed near Eq 10. It would prevent thinking that these coefficients are part of the wake model parameters. You could maybe place Figure 1c here to avoid scrolling through the document.

Figure 2: Min/max limits of x- and y-axes could be set at the same values to make the visual comparison easier.

Line 150: Please discuss the applicability of the power law extrapolation within waked flows.

Line 154: Why not use directly $U\_100m$ or other heights that are generally provided by atmospheric models? You are introducing a lot of uncertainties with Eq 15.

Figure 3: Explicit the name of the color bar.

**Section 3: Satellite observations and model data**

Line 159: SAR signal can be impacted by heavy rain.

Figure 4: For clarity purposes, it will help a lot to add when an extrapolation step is needed between U_10 and U_altitude and vice versa.

Line 173: Can you describe the steps where you calibrate the SAR data with SNAP? Is the starting point the SAR-derived wind speed from the L2 SAR product you get from https://scihub.copernicus.eu (which does not exist anymore, please change the source of your SAR data in the "Data availability" paragraph), or do you work directly with the L1 SAR product with SNAP?  Is SNAP-derived step depicted as the green box in your Figure 4? I am not sure to understand the purpose of using the SNAP tool to get NRCS. Can you quantify the uncertainty at each step of your procedure? Indeed, you could directly use L2-OCN products to have a validated NRCS and the associated *owiquality* flag, thanks to the CAL/VAL Sentinel-1 MPC.

Line 177: I understand that the 200-m resolution concerns the NRCS field and not the SAR-derived wind field. Please explain this.

Line 180: Please change NCRS into NRCS.

Line 181: Please rephrase the "radar signals from turbines". It is the interferences induced by operating turbines that are disturbing the SAR signal.

Line 182: Please define STDV.

Line 182: The result of this filtering step can be seen in Figure 8.a. Please discuss the results, especially in the left boundary of the domain.

Line 183: Where do the environmental conditions depicted in Figure 2a and Table A1 come from? The atmospheric model? The FINO1 platform? The SST may come from OSTIA, but T_2m? How is computed the average wind speed displayed in Table A1? Is it a neutral wind speed? Does ws come from CMOD5.N?

Line 191: In Figure 2b, you plot the wind speed from DWD: if you are not plotting a neutral component, how is it comparable to the neutral U_10m that comes from CMOD5.N?

Line 193: Please correct "a reasonable representations" with "a reasonable representation". Discuss the "reasonable" as it does not seem to be really the case.

Figure 5: Please define the symbols in the legend of the figure (with explanations in Line 275).

**Section 4. Inverse Modelling**

In general, it will help a lot the understanding of the methodology if you recall at each step the height of the wind speed and when you need to extrapolate between 10m and altitude. It seems that surface and altitude components of wind fields are mixed up in the following equations.

Line 199: GMF are derived from scatterometers/insitu collocations, and not SAR/insitu.

Line 201: Please discuss the impact of neutral atmospheric conditions on your work.

Line 213: Please introduce the BG notation. U and V are 10m components or related to the wind speed in altitude?

Line 225: Here, it seems that components at different heights are mixed up:

- From what I have understood in the previous parts, it is the wind speed in altitude (cf l155 "scaled according to Eq. 15") that is used as input of Wake2Sea. Hence, $\alpha_{wake}$ is supposedly related to a wind speed in altitude.
- $\beta_{j,k}$ is related to the height of (u,v) in Eq 16 and 17. I do not know if it is 10m or altitude
- NRCS must take as input a neutral wind speed at 10m as the CMOD5N GMF is used.

Could you check the consistency?

Line 225: If we put the previous mixing on heights aside, when implementing the difference between the NRCS derived from the simulation and the original SAR NRCS, how do you:

- Account for the time difference between the model and the SAR acquisition time? (-20 minutes to +17 minutes in Table A1)
- Retrieve the "neutral" component of the DWD wind speed?

By doing so, be aware that at this 10m altitude (if I assume the height correctly), you are qualifying the surface parameterisation model of the atmosphere model, and the quality of forcing data, rather than the quality of the atmosphere modelling itself. You will need to discuss whether the conclusions you find at 10m can directly be applied to hub heights.

Line 255: Please define TVD.

Figure 8:

- Explain MIN value in the legend.
- In 8b and 8c, some artefacts are visible in the middle of the domain due to some effects at the boundaries of the descending pass. Could you comment on this?

- In 8d, the red value in the standard deviation on the left side of the domain seems to be due to the noise we also find in the number of images in 8a. Is it due to the filters applied to the NRCS?

Line 279: Introduce notations ' and "

**Section 5: Inversion results**

Line 310: Discuss the robustness of 30 samples in terms of statistics. Does Figure 8 take into account the filtered NRCS?

Line 318: Discuss the situations which required stronger corrections. Is it related to the pixels in the vicinity of the boundary of the swath which were evicted with the NRCS filtering? Is it maybe related to some bias in your GMF implementation?

Figure 11: what is the value of DT in Figure c and d ?

Line 325: Is the analysis done without the correction of the background wind fields?

In general in section 5.1, you show aggregated results for one entire year. How does the model individually compare to SAR images outside of the "training" dataset?

Line 332: Please put "situation" in the plural form.

Line 340: It would be interesting to add FINO1 location in all figures.

Line 348: Why are you using a neutral log law to extrapolate your insitu data? Figure 2a shows that unstable conditions are more common in this site. Moreover, you used a power law extrapolation in your wake modelling. You should show some consistency.

Line 355: Please rephrase the sentence with the missing OWF wakes. Is a one-m/s bias in wind speed a standard value between unwaked and waked flows? Wake deficits can be much stronger. Please add references to support this order of magnitude.

Line 356: Mean absolute error would be an interesting metric. You could also investigate the comparison by wind speed ranges corresponding to wind turbine regimes, also related to the behaviour of the Ct curve.

Line 358-360: Please give some references supporting this theory.

Line 361: Discuss the 3%-deficit threshold

In general, other major reasons can also be discussed:

- The uncertainties related to the different uses of extrapolation laws, with log law in neutral conditions rather than unstable versus power law earlier in the methodology.

Dedicated methodologies exist to compare wind speed at different heights in the surface boundary layer (for instance a very recent work in https://wes.copernicus.org/articles/9/1727/2024/);

- The applicability of the Fitch parameterisation at 10m;
- Uncertainties related to the inversion model between wind speed and NRCS, and the assumption of neutral wind speed derived from CMOD5N;
- Uncertainties related to the optimisation methodology to derive the wake model;
- The representativity of the SAR dataset.

**Section 6: Theoretical considerations about Wake2Sea**

Line 370: Please rephrase "the shape of the wakes" into "the horizontal shape of the wakes".

Line 372: Recall the reference of the equation describing D.

Line 394: Please give references for the Gaussian shape profile.

Line 415: Recall that Figure 1.c shows that Ct=0.8 for a wind speed of 8 m/s. Here again, the thrust curve as the power curve of a wind turbine is a function of the wind speed encountered by the rotor in altitude. It can be the wind speed at hub height, or a rotor-averaged wind speed. The vertical profile of the wind speed in a wake is not well understood yet, but some studies also mention a Gaussian shape in vertical direction, whose lower part is more or less disturbed depending the nature of the ground. I am not sure that Eq 11 and 12 are able to account for this "Gaussian"-like vertical profile and the resulting behaviour between 10m and hub height.

Line 421: Please replace "suggest" with "suggests".

Section 7: Conclusions and Outlook

Line 432: Your explanation about the turbulence impact on the vertical extrapolation in waked flows is interesting but lacks scientific references. And may be of secondary importance compared to other reasons listed in my previous comments.

Line 436: Please discuss the marginal deviation from the standard formulations from the literature. Are you thinking about engineering wake models? If so, please add comparisons with your model, and add references to benchmarking studies with engineering wake models.

Line 442: Please explain how you obtained the 36% value. Discuss how you can derive your results in surface to altitude considering the limitations that have been highlighted, especially due to the major uncertainties in the vertical profiles in waked flows inside a wind farm and within a cluster of wind farms.

---

## Referee Comment (RC2)

**Review of "A semi-empirical model for near sea surface wind speed deficits downstream of offshore wind parks fitted to satellite synthetic aperture radar measurements" submitted for publication in Wind Energy Science.**

The presented manuscript introduces a two-dimensional model for the estimation of near-surface wind speed due to the impacts of wind-farm wakes. The model applies data assimilation to a set of empirical relations coupled with a physics-based model of the near-surface atmosphere (hence the name semi-empirical) using Satellite Aperture Radar (SAR) images as reference. Validations of the optimized model are presented against, for instance, recordings from the FINO-1 met mast. The purpose of the model is to have a computationally less expensive tool than three-dimensional atmospheric models to estimate near-surface conditions impacted by wind-farm wakes, which has direct application in oceanography.

The manuscript clearly consists of two main parts: the physics model of the atmosphere, named "Wake2Sea", and the data assimilation technique. This review will mostly focus on the first part: the physics model, but suggestions to other parts will be included.

First, some general notes about the manuscript are presented. The arrangement of the figures and their corresponding text needs to be revised. Figures were typically a few pages ahead of their corresponding text, which makes it very hard for the reader not to lose focus while going back and forth. For instance, Fig. 5 is in page 10 and is first mentioned six pages after (page 16). Additionally, the mathematical variables used in both main parts of the manuscript are not well defined, which adds further difficulties to understand. The authors should consider a thorough description of all used variables. Additionally, the manuscript can benefit from polishing the text for better readability.

I have some concerns about the assumptions made to derive the physics model of the near-surface atmosphere. These concerns are detailed below. However, I wanted here to emphasize a critical point regarding models that fit to either high fidelity simulations or to measurements. When comparing fitted models to reference data, matching the reference data is not an indication of the solidness of the adopted physical model. A flawed physics model, if fitted properly, can match the reference data potentially leading to an overestimation of the model's robustness. Applying the fitted model to conditions that were not included in its optimization process is typically a better test for the model's accuracy. The Wake2Sea model was fitted to and tested against SAR images of the North Sea only, which does not actually measure the accuracy of the underlying physics assumptions. The same analogy holds for the comparison against the recordings of the FINO-1 mast, which is situated in the North Sea. This indicates that the near-surface wind conditions at the FINO-1 site were already included in the optimization of the Wake2Sea model. Hence, such comparison mainly examines the accuracy of the expression used to relate the wind speed at 10 m above the surface (used in the optimization process) to the wind speed at the recording height, rather than testing the physics accuracy of the Wake2Sea model.

Based on this introduction and on the comments listed below, I recommend **rejection** in its current form, as the manuscript does not yet meet the publication standards of Wind Energy Science.

**The Wake2Sea model**

Momentum extraction by the turbines was modelled using Fitch's parameterization. However, Fitch's parameterization represents the turbines as not only sinks of momentum, but also sources of turbulence. The role of turbine-induced turbulence remains unclear, given that its impact on the flow, particularly near-surface fluxes, is not negligible. If turbine-induced turbulence was not considered, it was not discussed what implications this assumption may have on the model's accuracy.

The derivation of Eq. 2 is unclear and would benefit from further clarification. The first assumption (setting $U_- \approx 0$) can be to some extent understood if the authors intended to say that the wind speed at the bottom of the considered atmospheric layer is much smaller than wind speeds at higher altitudes. However, setting $U = U_+$ needs further elaboration. While not mentioned, can I assume here that the authors intended to say that the vertical profile of wind speed becomes uniform starting from the middle of the considered layer, and hence does not change with higher altitude? If this is the case, which can roughly be pictured to represent an atmospheric boundary layer where the wind profile reaches geostrophic values, then the height of the considered layer should be at least twice the atmospheric boundary layer thickness. However, later in the article, the authors mention that the thickness of this layer is taken to be 200 m, which is much less than a typical atmospheric boundary layer height. The authors should elaborate more on the consistency of their assumptions regarding this part, particularly the consistency of the assumptions in Eq. 2 and the selection of the atmospheric layer thickness.

Line 94: "Changes in the pressure field introduced by OWFs are not considered". Can this be supported from the literature by showing the typical distances downstream of a wind farm after which the farm's impact on the pressure field becomes negligible?

Line 95: "The advection term for the deficit includes higher order terms, which were omitted to keep the numerical treatment simple". It would be helpful if the authors first presented the higher-order terms and then provided justification for their omission. Terms can be neglected when they are considerably smaller than other dominant terms in an equation. This is typically done analytically using an order of magnitude analysis or numerically using a budget analysis of the considered equation. None of these approaches was included.

The authors assumed that the considered turbines in the North Sea area have the same thrust coefficient. I understand that finding such data for individual turbine types is not easy. However, neglecting such variations may impact the accuracy and generalizability of the Wake2Sea model.

Line 141: Setting the CFL number to unity is not a universal stability condition. While it holds for simple partial differential equations (e.g. a heat equation), for more complex PDEs this condition does not necessarily guarantee numerical stability. Can the authors comment on their choice of setting the CFL number to one? Some support from the literature would benefit.

Line 394: "If we assume that the across wake profile has a Gaussian shape at distance x = 0 km from the wind farm". This assumption needs further justification. Wind-farm wakes do not become self-similar and follow a Gaussian profile just behind the wind farm.

Line 438: "The sink term also showed a slight dependency on the deficit itself with lower diffusion at higher deficits". It is unclear how the sink term, which represents the momentum

extracted by the wind farm would be dependent on one of its results which is the wind-speed deficit. Can the authors elaborate more?

**Minor comments**

The following are minor comments to be considered by the authors if they please.

- Fig 1a, b: Consider using a discrete colour bar instead, as it may enhance readability.
- The introduction section frequently refers to things that are yet to be introduced later in the manuscript. This may reduce the clarity for readers unfamiliar with the material.
- Lines 158-172: Can this be summarized? This part is very detailed, and not all these details are relevant to the main purpose of the manuscript.
- Lines 178-182: Presenting this section as bullet points may improve readability.
- In the caption of table A1, please briefly mention what is $\Delta T$ and $U_{10}$. Are they spatial averaged values or are they taken at the FINO-1 site as in Fig. 2?

**Suggestions**

I have a few suggestions to the authors to consider if they please.

- The underlying physics assumptions of the presented model needs to be thoroughly revised.
- A clear mention of the limitations of the proposed model would help the reader understand when it is suitable to use the model.
- Consider validating the model with SAR images for other sites than the North Sea.
- The impact of turbine-induced turbulence should be accounted for.

---

## Author Comment (AC1)

Manuscript No: wes-2025-59

MS type: Research article

Title: A semi-empirical model for near sea surface wind speed deficits downstream of offshore wind parks fitted to satellite synthetic aperture radar measurements

Journal:  Wind Energy Science

**Answers to the reviewers**

*General comments to all two reviewers*

*We would like to thank all reviewers for their careful reading of the manuscript and for their fair and constructive remarks. Please find below the details on how the specific points raised by the reviewers were addressed. Reviewers' comments are in black and our answers are in blue italic.*

*The responses refer to the old numbering to avoid confusion. Also, comments about figures refer to the original numbering unless indicated otherwise.*

*Beside the current file for the answers to the reviewers, two additional pdf files are attached. One refers to the new pdf version of the paper after taking into account the reviewers' comments. The last pdf file is the file that highlights all the changes that were made to the first submitted version (difference between the revised version and old version of the paper). In the latter file, the changes are characterised by the underlined blue text, which is the new correct text and the strikethrough text in red, which is the text that has been removed.*

**General Comment on Revisions**

*As a general comment, we would like to emphasize that the core of this study is the inversion of the wake model trying to condense the complex information from SAR data into a compact format and providing a simple tool to access this information. We would also like to stress again that the study represents the first attempt to make progress in this direction. We believe that the fact that no studies addressing this issue exist so far is because of the technical challenges and not because of a lack of interest. With growing complexity of the forward model the treatment of the inversion problem becomes exponentially more difficult, and we therefore had to find a compromise between physical consistency and practicality. It is obvious that a model of this relatively low level of complexity misses certain aspects of the dynamical processes, and this will still be the case even if we add an additional prognostic equation for turbulence or extend the model in other ways. We experimented a lot with different types of parameterisations including versions with stronger*

*physical consistency and as a matter of fact the presented version outperformed alternative formulations by far.*

**Reviewer N#1**

The work described in "A semi-empirical model for near surface wind speed deficits downstream of offshore wind parks fitted to satellite synthetic aperture radar measurements" is of great interest as it relies on large-scale satellite observations to improve the accuracy of wake modelling in datasets from numerical atmospheric models. SAR imagery is indeed highly valuable information about the surface wind field in coastal and offshore locations and deserves to be better known in the offshore wind community.

While the scientific aspect and the innovative source of data are of great interest to the research community, the paper lacks consistency in the final application and the description of the methodology. Indeed, the authors emphasize that the final application is related to the improvement of surface wind speeds for the forcing of ocean models. However, the context of ocean modelling is rather poorly described, the focus being made by the authors on the wind speed in altitude and offshore wind energy in general.

*We thank the reviewer for the thoughtful and constructive feedback, and for highlighting the scientific value of using SAR data to improve wind wake modelling.*

*We acknowledge the reviewer's concern regarding the link between wake modelling and its intended application in ocean modelling. While Wake2Sea is indeed designed to enhance atmospheric forcing for ocean simulations, demonstrating this application lies beyond the scope of the present study and will be addressed in forthcoming work.*

*To clarify the scope, we have revised the manuscript to emphasize that this paper focuses on the development, calibration, and validation of the Wake2Sea model as a standalone tool for simulating near-surface wind deficits using satellite observations. We also now refer to Christiansen et al. (2022), who applied a simpler SAR-based empirical model in an ocean modelling context. In contrast, Wake2Sea incorporates additional physical processes, such as atmospheric stability effects and turbine-specific thrust modelling, offering a more robust basis for future ocean application. We have furthermore added a statement about the fact that ocean modelers both in the ocean circulation and ocean wave modelling community use U10, i.e. the wind speed 10 m height above the surface as a standard parameter for the surface forcing.*

*We have also explained the relationship between the layer-averaged dynamical equation and the wind speed deficit near the surface in more detail.*

Secondly, the description of the global methodology needs to be reviewed with particular care on the uniformisation of notations, precise mention of the considered height at each step of the process. Mixing parameterisations and concepts at different altitudes makes it complicated to follow the methodology and makes the whole process questionable with different extrapolations concepts in waked flows that are not properly discussed. The authors try to implement a rather simplistic approach based on some parameters to account for the change in altitude into their model. Indeed, the use of Fitch parameterisation and Ct requires a wind speed in altitude, the input of their model is a 10-m wind speed which is scaled up to address a wind field in altitude. But the output of the model is a surface wind field with inclusion of a wake. The model is trained/optimised on the difference between backscattered signals which are supposed to represent the surface wind field. A lot of uncertainties due to the change in altitude embarked in the whole methodology makes the conclusions of the author (i.e. an absence of turbulence modelling in their model to extrapolate properly insitu data which measurement heights are higher than 10m) hard to validate.

*We thank the reviewer for this detailed and insightful comment. We agree that the vertical consistency of wind speed treatment and clear distinction of height references throughout the methodology are essential.*

*We acknowledge that the current version of the manuscript could be improved in terms of notation consistency and clarity about the considered altitudes at each stage. We have carefully revised the methodology section to clearly indicate the reference height (10 m, hub height, or other) used in each part of the modelling chain and to standardize notation throughout the paper.*

*We have revised the text to make it clearer that the prognostic variable D refers to the layer-averaged wind speed. This is also consistent with the formulation used in the Fitch parameterization. The connection between the layer-averaged quantities and the surface parameters are indeed simplified (see general comment), but with the revisions the approach is at least now more transparent.*

Thirdly, the authors use satellite imagery. The description of this remote sensing technique, as well as the resulting data, is very interesting and brings a new horizon to the offshore wind community. However, the offshore wind community is not familiar with this data, and it is important to demonstrate that the SAR-derived wind field does not bring additional uncertainties. I advise the authors to properly describe their methodology with the SNAP tool distributed by ESA and to explain why they do not use the L2 OCN products that include wind speed, NRCS and

quality flags, that are provided and validated by the CALVAL of the Sentinel-1's Mission Performance Center.

*We thank the reviewer for the comment. We agree that transparency is key to ensuring confidence in SAR-derived wind fields. To address this, we have revised the methodology section to more clearly describe our SAR processing approach, including the steps involved in NRCS calibration, wind inversion using Geophysical Model Functions (GMFs), and incorporation of wind direction information.*

*While the Sentinel-1 Level-2 OCN products provide wind speed, while the L2 products provide wind speed, NRCS, and quality flags and are validated by the Sentinel-1 Mission Performance Center, we opted for a custom processing chain in this study for the following reasons:*

- *the custom approach offers higher spatial resolution and greater flexibility, which is critical for accurately resolving the fine-scale wake structures downstream of offshore wind farms. The L2 products are rather coarser.*
- *the custom approach provides direct control over the wind inversion process, allowing the use of auxiliary wind direction inputs (here DWD). The inputs from DWD offer higher spatial resolution than the ECMWF wind fields used in the ESA L2 products. Wind direction is actually critical for retrieving accurate wind speed. Wind fields derived from the custom SAR processing show good agreement with DWD data, even at high spatial resolution. This confirms the reliability of the retrievals and underscores the value of using regionally optimized input data.*
- *while the L2 wind fields rely on standard geophysical model function (GMF) inversions, they offer limited transparency and tunability.*
- *The L2 product (which is the result of an inversion procedure itself) requires external information about wind direction, which is not necessarily consistent with the DWD model data used in this study. This creates a lot of additional inconsistencies in the inversion process. In general, it is preferable to use lower processing levels of the observation data in the inversion process and to use respective observation operators to simulate these data.*

*The section is revised accordingly: "The SAR data were radiometrically calibrated to obtain the Normalized Radar Cross Section (NRCS) using the SNAP (Zuhlke et al., 2015) software made available by the European Space Agency (ESA). This radiometric calibration ensures that the pixel values represent physically meaningful backscatter coefficients independent of acquisition geometry. After calibration, the images were terrain-corrected. To reduce speckle noise (Kerbaol, 1997), the SAR images were smoothed down to 200 m grid resolution. The NRCS is a dimensionless quantity, which describes the intensity of the radar return and it is often expressed in decibels (dB) values. For assimilation into the inversion framework we used linear units of NRCS, while dB values*

*are shown only for better visualisation in selected figures. .... We emphasize that we did not use the Sentinel-1 Level-2 Ocean (OCN) products, even though they provide wind speed, NRCS, and quality flags. Our assimilation framework requires continuous calibrated NRCS values as direct input, not preprocessed wind vectors. Moreover, the OCN products are delivred at coarser resolution and do not allow detailed control of calibration and filtering (e.g. removal of ship signatures, turbine returns, and shallow-water artefacts), which is essential for reliable wake inversion."*

I therefore recommend a major revision of this paper.

Detailed comments on the document:

A general comment on the position of the figures in the article: it will help to have the figures closer to the paragraph mentioning and explaining the figures.

*We thank the reviewer for the comment. We have adjusted the placement of the figures so that they now appear closer to the corresponding paragraphs in which they are introduced in order to improve the readability of the manuscript.*

Section 1: Introduction

Line 28: Please replace "industry" with "engineering".

*Done.*

Line 54: You could emphasize that the addition of the wakes is done a-posteriori to existing surface datasets.

*We thank the reviewer for this helpful suggestion. We agree that it is important to clarify that the integration of OWF wakes using the Wake2Sea model is performed a posteriori.*

*We have revised the corresponding sentence and improve clarity for the reader:*

*"The proposed semi-empirical model can be applied a posteriori to existing atmospheric model datasets to incorporate OWF wake effects. Many of these data …"*

*"The Wake2Sea model enables the inclusion of OWF wakes a posteriori into existing atmospheric model data sets at significantly lower computational costs compared to complete re-runs of full blown 3D atmospheric models."*

Line 56: The application of the tool is in contradiction with Lines 64-65. This sentence needs to be discussed and rephrased. The OWF context generally means wind speed or yield assessment in altitude, or structural analysis (from floaters/foundations up to the upper tip of the blades).

*We thank the reviewer for the comment.*

*We have clarified in the revised manuscript that the Wake2Sea model targets the simulation of near-surface wind deficits (e.g., at 10 m height) and is not designed for wind power assessments at turbine altitude. The intended application of the model is to enhance the representation of offshore wind farm wake effects in atmospheric datasets, particularly for use in oceanographic and environmental modelling. The relevant sentence has been rephrased accordingly to reflect this scope more accurately:*

*"Many of these data sets, like ERA5 (Hersbach et al., 2020), are intensely used as references by the scientific community and the proposed tool can massively enhance the applicability of these data for studying near-surface wake effects in the offshore wind farm context."*

Figure 1: In general, color bars may be easier to understand with the title next to it.

We thank the reviewer for this practical comment.  We have updated Figure 1 so that the color bar includes the title next to it, which improves clarity and readability.1

Figure 1c: Ct curve: wind speed at hub height or rotor-averaged wind speed. Is Ct extended for u_hub > 25m/s with a stall to a zero-Ct value?

*We thank the reviewer for the comment. The Ct curve in the figure is plotted as a function of hub-height wind speed, consistent with the formulation used in the Fitch parameterization and the implementation in Siedersleben et al. (2018), which our model builds upon.*

*Regarding high wind speeds beyond 25 m/s: we do not apply a stall or drop Ct to zero.  behavior or set Ct = 0 for u > 25 m/s. Instead, the Ct value is held constant at the last defined value of the smoothed thrust curve. This avoids introducing discontinuities and is consistent with typical practice in wake modelling when operating near or beyond the turbine cut-out wind speed.*

*"…c) Smoothed version of the turbine thrust CT-curve introduced in Siedersleben et al. (2018), shown as a function of hub-height wind speed."*

Figure 1d: I am not sure to understand the relevance of this graph. Would it be more interesting to locate the power of the turbines on the map as in a and b? Please describe or introduce dZ notation.

*We thank the reviewer for the useful comment.*

*The purpose of Figure 1d is to illustrate the layer of thickness dZ used in the prognostic equation for the vertically averaged quantities. We have modified the caption accordingly.*

*"Figure1: …d) Scatter plot of lower vs. upper rotor tip height for offshore turbines in January 2023, colored by rated turbine power (in MW). The vertical bracket labeled dZ. represents the vertical layer, which is used for the averaging of the prognostic variables in the two-dimensional wake model."*

Line 59: Discuss the choice of the SAR scenes and how only 30 is relevant to cover all situations.

*We thank the reviewer for the comment. As all SAR scenes have to be inverted simultaneously (because the wake model parameters have to be the same for all scenes), there is a limitation with regard to the computational resources and the number of images that can be used in the inversion process. To keep the computation time within feasible limits we also had to parallelize the inversion procedure of each image resulting in the use of 480 processors.*

*Figure 2 shows that we cover typical wind speed and stability situations observed in the German Bight. The Figure was revised to have a better match of the axis limits in a) and b).*

Line 61: The reference to the Fitch parameterisation needs to be added.

*We thank the reviewer for the comment. The reference to the Fitch parameterisation has been added: "The Fitch paramerisations (Fitch etal., 2012) is used to include OWF properties, e.g. the …"*

Line 61: Please uniformize the notation of the thrust coefficient, Ct or CT, throughout the paper.

We thank the reviewer for pointing this out. We have now uniformized the notation for the thrust coefficient throughout the manuscript and consistently use CT in all instances.

*"…c) Smoothed version of the turbine thrust CT-curve …"*

*"To ensure …CT curve on different … "default" CT curve …"*

Line 66: Explicit the height of the "near surface wind". Is it 10m?

*We thank the reviewer for the comment.  The "near surface wind" refers to wind speed at 10 m height. We have revised the sentence accordingly to make the reference height explicit:*

*"… speed deficits within wind farms for the 10 m near surface wind compared…"*

Section 2: Wake2Sea

In general, mathematical formulations need to be uniformized throughout the paper. Notations and parameters need to be introduced and clearly defined at each step.

*We thank the reviewer for the comment.  We have revised the manuscript in different places accordingly. In particular we have*

1) *added an explanation that the quantities in eq. 1 refer to averages over the layer dZ*
2) *In eq. 9 we have replaced "alpha" by "x", to make it clearer that this is just an independent variable used to define the function and to avoid confusion with the control vector.*

When dealing with Ct and the wind speed U, it is not always clear at which height the equations are considered (10m or hub height). The applicability of Fitch parameterisation at 10m with Ct defined with U_10m needs to be strongly discussed.

*We thank the reviewer for the comment.  The prognostic variable U refers to the mean over the entire layer dZ and the Fitch term also refers to this layer as in*

> *Fitch, A.C., Olson, J.B., Lundquist, J.K., Dudhia, J., Gupta, A.K., Michalakes, J., Barstad, I., 2012. Local and mesoscale impacts of wind farms as parameterized in a mesoscale NWP model. Monthly Weather Review 140, 3017–3038.*

*The difference is of course that the layer used in our study is much thicker than the layers typically used in 3D atmospheric models, but formally the approach is the same.  The idea is that the mean over the lowest two hundred meters is a reasonable estimate for the wind speed at hub height (approx. 100 m, i.e. roughly the centre of the layer).*

For clarity, it would help to:

- Mention at the beginning of Section 2 that the eight constant parameters to be fitted will be called \alpha_i€(1-8) in the following equations;
  *We thank the reviewer for the comment.* We added the following after eq. 2

  "Including the additional components of the model described in the following, Wake2Sea contains eight uncertain parameters $\alpha_1, \alpha_2, \ldots, \alpha_8$, which are estimated as part of the inversion process."

- Recall the final system of equations in Line 130.

  *We thank the reviewer for the comment. We have added the following part to remind the reader about the meaing of the different equations:*

  *"The prognostic equation for the deficit D eq. 7, the expression for the vertical deficit diffusion eq.10, the correction function for the Fitch parameterisation eq. 13, and the simplified expression relating layer averaged wind deficits to surface deficits eq. 14 represent the semi-empirical wake model Wake2Sea that ...."*

Line 74: Rephrase "bottom layer of the atm above the ocean surface".

*We thank the reviewer for the comment. We rephrased this as*

  *"... for a layer of thickness dZ of the atmosphere above the ocean surface ..."*

Line 75: Is U at 10m? Within the surface layer? Within the whole MABL?

*We thank the reviewer for the comment. We have added an explanation after eq. 1 that the quantities in this equation refer to averages over the vertical layer of thickness dZ.*

Line 80: Define \Chi = \nu_v/dZ² as the vertical diffusion parameter.

*We thank the reviewer for the comment. After some consideration we kept the present text, because the suggested modification could lead some readers to believe that \xi is the vertical diffusion coefficient (but \xi has the unit 1/s).*

Line 81: Uniformize upper or lower case for u+/u-, and U in Lines 75 and 81. Line 82: Please recall the Fitch parameterisation (its mathematical formulation) as it would make it easier to understand the following equations.

*We thank the reviewer for the comment.  We changes u+/- to U+/-*

*We added the following sentence to make the link with the underlying Fitch model clearer:*

> *"Eq. 5 corresponds to eq. 8 in Fitch et al. (2012)."*

Lines 94 and 95: Please discuss the simplifications with order of magnitude.

We added the following text:

"The major impacts on the pressure field are very local with distances from the wind farm of the order of the wind farm size (Smith, 2023). As we have not observed pressure related phenomena on SAR images, e.g. blockage, and because an inclusion would lead to a much higher complexity of the model we decided to not include this effect in the first version. *The approximation of the advection term in particular means that the deficit is advected with the unperturbed background wind and not with the reduced wind speed, i.e. this can lead to errors in the advection of the order of 10%."*

Smith, R., 2023. The wind farm pressure field. Wind Energy Science Discussions 2023, 1–14. https://doi.org/10.5194/wes-9-253-2024

Line 109: Define \DeltaT: is it T_2m – SST?

*We thank the reviewer for pointing this out. Yes, the term \DeltaT refers to the air-sea temperature difference, where T2m is the 2 m air temperature and SST is the sea surface temperature . We have now explicitly defined this in the manuscript at its first occurrence.*

*"…with the air/sea temperature difference ΔT (i.e., ΔT =T_2m – SST) and a differentiable function…"*

Line 110: Introduce \alpha.

*We thank the reviewer for the comment.  We changed alpha to x to make it clearer that this parameter is just the independent variable needed to define the function and to avoid confusion with the control vector.*

Line 121: Ct(U) with U at hub height. I am not sure that Ct(U downscaled to 10m) means anything.

*We thank the reviewer for the comment. We shifted the paragraph around equation 14 in front of the introduction of the Fitch parameterisation around equation 3.  In doing so it should become clearer that U refers to the layer average, which should be reasonably close to the wind speed at 100 m, because the layer thickness is 200 m.*

Line 126: Please discuss this assumption, especially the applicability of the Fitch parametrization at 10m.

*We thank the reviewer for the comment. Please see the response to the previous comment (comment on line 121).*

*The original Fitch parameterization for the momentum sink is designed for use at any layer that intersects with the rotor disc areas. However, since our model aims to simulate near-surface wind deficits (at 10 m height) to match SAR-derived wind fields, an empirical adjustment is required to translate the modeled hub-height wake deficit to the 10 m level. The eq. 11 was introduced as a simple correction scheme to account for vertical effects (such as changes in wake structure, turbulence dissipation, and atmospheric stability that influence the wind profile from hub height to the surface).*

Line 136: dZ is related to the height of the MABL? Is it constant for all stability conditions? A recall of the standard Fitch parametrization would be easier to understand if recalled in Line 82 to introduce the concepts and notations, especially of dZ and the "bottom layer" which seems to be a modelling concept rather than a physical part of the MABL.

*We thank the reviewer for the comment. Yes, dZ is in fact related to computational aspects and it is not meant to be a characteristic length scale of the ABL. dZ is the layer thickness used to define the vertically averaged quantities. The main side condition for the definition of dZ was that it should include the major parts of all rotor disc areas to make eq. 5 consistent with the Fitch model. We already included more detailed information which equation exactly in Fitch et al. [2012] we are using in our study in the paragraph following eq. 5.*

Table 1: The table could be placed near Eq 10. It would prevent thinking that these coefficients are part of the wake model parameters. You could maybe place Figure 1c here to avoid scrolling through the document.

*We thank the reviewer for this helpful suggestion. We agree that positioning Table 1 closer to Eq. 10 now (Eq. 12) would help avoid any confusion. We have moved the table accordingly.*

Figure 2: Min/max limits of x- and y-axes could be set at the same values to make the visual comparison easier.

*We thank the reviewer for the comment. We adjusted the axis limits in Figure 2 a,b to match exactly.*

Line 150: Please discuss the applicability of the power law extrapolation within waked flows.

*We thank the reviewer for the comment.*

*We have added the following text after eq. 4:*

*"We are aware that the approximation used in eq. 4 is very crude, however there is still overall uncertainty concerning model representations of the vertical ABL structure within wind farm wakes. Our decision to use this simplification was also driven by the necessity to keep the model simple to ensure a stable inversion of the satellite data."*

Line 154: Why not use directly U_100m or other heights that are generally provided by atmospheric models? You are introducing a lot of uncertainties with Eq. 15.

*We thank the reviewer for the comment. As explained in the introduction, one motivation for the study is to provide a simple tool for oceanographers to include offshore wind farm wakes in their ocean forcing. The standard parameter used by oceanographers for the mechanical meteo forcing is the wind u10 at 10 m height and this is what they usually have available in their data bases. We*

*were at some point considering to use ERA5 data for our study (which of course includes U100), but the spatial resolution of these data is significantly coarser than what we have from DWD.*

Figure 3: Explicit the name of the color bar.

*We thank the reviewer for the comment. The colorbar refers to the dimensionless spline values and we added a respective explanation in the caption.*

**Reviewer N#2**

The presented manuscript introduces a two-dimensional model for the estimation of near-surface wind speed due to the impacts of wind-farm wakes. The model applies data assimilation to a set of empirical relations coupled with a physics-based model of the near-surface atmosphere (hence the name semi-empirical) using Satellite Aperture Radar (SAR) images as reference. Validations of the optimized model are presented against, for instance, recordings from the FINO-1 met mast. The purpose of the model is to have a computationally less expensive tool than three-dimensional atmospheric models to estimate near-surface conditions impacted by wind-farm wakes, which has direct application in oceanography.

The manuscript clearly consists of two main parts: the physics model of the atmosphere, named "Wake2Sea", and the data assimilation technique. This review will mostly focus on the first part: the physics model, but suggestions to other parts will be included.

First, some general notes about the manuscript are presented. The arrangement of the figures and their corresponding text needs to be revised. Figures were typically a few pages ahead of their corresponding text, which makes it very hard for the reader not to lose focus while going back and forth. For instance, Fig. 5 is in page 10 and is first mentioned six pages after (page 16). Additionally, the mathematical variables used in both main parts of the manuscript are not well defined, which adds further difficulties to understand. The authors should consider a thorough

description of all used variables. Additionally, the manuscript can benefit from polishing the text for better readability.

I have some concerns about the assumptions made to derive the physics model of the near-surface atmosphere. These concerns are detailed below. However, I wanted here to emphasize a critical point regarding models that fit to either high fidelity simulations or to measurements. When comparing fitted models to reference data, matching the reference data is not an indication of the solidness of the adopted physical model. A flawed physics model, if fitted properly, can match the reference data potentially leading to an overestimation of the model's robustness. Applying the fitted model to conditions that were not included in its optimization process is typically a better test for the model's accuracy. The Wake2Sea model was fitted to and tested against SAR images of the North Sea only, which does not actually measure the accuracy of the underlying physics assumptions. The same analogy holds for the comparison against the recordings of the FINO-1 mast, which is situated in the North Sea. This indicates that the near-surface wind conditions at the FINO-1 site were already included in the optimization of the Wake2Sea model. Hence, such comparison mainly examines the accuracy of the expression used to relate the wind speed at 10 m above the surface (used in the optimization process) to the wind speed at the recording height, rather than testing the physics accuracy of the Wake2Sea model.

*We thank the reviewer for the comment. We adjusted the positioning of the figures and placed them closer to the referencing text. We added more details on the definition of the various variables. In response to Reviewer 1 we explained the treatment of the vertical dimension more thoroughly.*

*With regard to physical consistency of the model we would like to repeat the comment already made in the general introduction: We would like to emphasize that the core of this study is the inversion of the wake model trying to condense the complex information from SAR data into a compact format and providing a simple tool to access this information. We would also like to stress again that the study represents the first attempt to make progress in this direction. We believe that the fact that no studies addressing this issue exist so far is because of the technical challenges and not because of a lack of interest. With growing complexity of the forward model the treatment of the inversion problem becomes exponentially more difficult, and we therefore had to find a compromise between physical consistency and practicality. It is obvious that a model of this relatively low level of complexity misses certain aspects of the dynamical processes, and this would still be the case even if we added an additional prognostic equation for turbulence or other elements. We experimented a lot with different types of parameterisations including versions with stronger physical consistency and as a matter of fact the presented version outperformed alternative formulations by far.*

*We understand that there is a strong tendency to use machine learning approaches to develop tools as discussed in our study. We decided not to go that route to be able to include first order physical principles in the model, to keep the number of model parameters small and the model transparent. As explained above the price we had to pay was strong simplifications in several components of the model.*

*With regard to the fitting procedure it must be emphasized that the information content contained in the 30 SAR images is much higher than the information represented by the eight control parameters for the wake model, because each SAR contains various wind farms and corresponding wake features. This means that there is no risk that the inversion is overfitting, which would certainly have been an issue if we had applied machine learning techniques.*

*To emphasize that there are of course limitations regarding the general applicability of the model we revised the title of the paper as follows:*

*"A semi-empirical model for near sea surface wind speed deficits downstream of offshore wind parks in the German Bight fitted to satellite synthetic aperture radar measurements"*

Based on this introduction and on the comments listed below, I recommend rejection in its current form, as the manuscript does not yet meet the publication standards of Wind Energy Science.

**The Wake2Sea model**

Momentum extraction by the turbines was modelled using Fitch's parameterization. However, Fitch's parameterization represents the turbines as not only sinks of momentum, but also sources of turbulence. The role of turbine-induced turbulence remains unclear, given that its impact on the flow, particularly near-surface fluxes, is not negligible. If turbine-induced turbulence was not considered, it was not discussed what implications this assumption may have on the model's accuracy.

*We thank the reviewer for the comment.*

*As explained in the general comments we had to find a balance between the physical consistency and simplicity of the model in the context of the inversion problem. Turbulence is implicitly included in the model through the dependence of the vertical diffusion on the deficit and the stability. A thorough comparison between fully-fledged atmospheric models and SAR data does so far not exist and it is yet unclear if the turbulence models, e.g. used in mesoscale models are*

*able to reproduce the signatures found on SAR scenes. This was another motivation to start with a simple model and to check which complexity is needed to reproduce the major SAR features.*

The derivation of Eq. 2 is unclear and would benefit from further clarification. The first assumption (setting $U-≈0$) can be to some extent understood if the authors intended to say that the wind speed at the bottom of the considered atmospheric layer is much smaller than wind speeds at higher altitudes. However, setting $U=U+$ needs further elaboration. While not mentioned, can I assume here that the authors intended to say that the vertical profile of wind speed becomes uniform starting from the middle of the considered layer, and hence does not change with higher altitude? If this is the case, which can roughly be pictured to represent an atmospheric boundary layer where the wind profile reaches geostrophic values, then the height of the considered layer should be at least twice the atmospheric boundary layer thickness. However, later in the article, the authors mention that the thickness of this layer is taken to be 200 m, which is much less than a typical atmospheric boundary layer height. The authors should elaborate more on the consistency of their assumptions regarding this part, particularly the consistency of the assumptions in Eq. 2 and the selection of the atmospheric layer thickness.

*We thank the reviewer for the comment.*

*It is true that the choice of dZ=200 m was based on the necessity to include the rotor disc areas, but at the same time to keep it small so that the vertical average is still representative for what is going on in the layer. We are aware that the approximation U+ = U is very crude and we changed this to U+ = alpha_chi U with a stability dependent parameter alpha_chi. The resulting functional form is still the same and as explained later on, chi is indeed stability dependent in our model (eq. 10). The text was adjusted accordingly.*

Line 94: "Changes in the pressure field introduced by OWFs are not considered". Can this be supported from the literature by showing the typical distances downstream of a wind farm after which the farm's impact on the pressure field becomes negligible?

*We thank the reviewer for the comment. We added the following text after the bullet point list following eq. 7*

*"The major impacts on the pressure field are very local with distances from the wind farm of the order of the wind farm size (Smith, 2023). As we have not observed pressure related phenomena*

on SAR images, e.g. blockage, and because an inclusion would lead to a much higher complexity of the model we decided to not include this effect in the first version."

Smith, R., 2023. The wind farm pressure field. Wind Energy Science Discussions 2023, 1–14. https://doi.org/10.5194/wes-9-253-2024

Line 95: "The advection term for the deficit includes higher order terms, which were omitted to keep the numerical treatment simple". It would be helpful if the authors first presented the higher-order terms and then provided justification for their omission. Terms can be neglected when they are considerably smaller than other dominant terms in an equation. This is typically done analytically using an order of magnitude analysis or numerically using a budget analysis of the considered equation. None of these approaches was included.

*We thank the reviewer for the comment.*

*The complete expression for the advection term would be*

$$D(2 - D)\nabla u + u(1 - D)\nabla D$$

*The first term was omitted because we neglected variations of the background wind field on the wake scale. The second term was simplified because we neglected the reduction of the advection speed due to the deficit. If we had kept the term $D \nabla D$, we would basically have created turbulence in the deficit, which creates substantial challenges in the inversion.*

The authors assumed that the considered turbines in the North Sea area have the same thrust coefficient. I understand that finding such data for individual turbine types is not easy. However, neglecting such variations may impact the accuracy and generalizability of the Wake2Sea model.

*We thank the reviewer for the comment.*

*We agree with your statement, but it is impossible for us to get access to reliable information about Ct curves for the majority of the installations. We believe that speculation about differences in the thrust curves would add even more inaccuracies.*

Line 141: Setting the CFL number to unity is not a universal stability condition. While it holds for simple partial differential equations (e.g. a heat equation), for more complex PDEs this condition does not necessarily guarantee numerical stability. Can the authors comment on their choice of setting the CFL number to one? Some support from the literature would benefit.

*We thank the reviewer for the comment.*

*As the estimated time step referred to 25 m/s which is significantly larger than the wind speeds observed in the tuning data set, we are effectively using a CFL number below 1 in the inversion. The time step is critical for the overall computational burden of the inversion as both the forward and the adjoint model have to be run many times and the chosen value for the time step turned out to be a good compromise between run time optimisation and numerical stability – in fact we did not observe any numerical instabilities in the inversion.*

Line 394: "If we assume that the across wake profile has a Gaussian shape at distance x = 0 km from the wind farm". This assumption needs further justification. Wind-farm wakes do not become self-similar and follow a Gaussian profile just behind the wind farm.

*We thank the reviewer for the comment.*

*There is in fact very little known about the shape of the across wake profiles and it must be assumed that it depends on various environmental and installation parameters. As often done in such situations we decided to use a simple functional form (the most classical one that one could think of) which lends itself for analytical investigations.*

Line 438: "The sink term also showed a slight dependency on the deficit itself with lower diffusion at higher deficits". It is unclear how the sink term, which represents the momentum extracted by the wind farm would be dependent on one of its results which is the wind-speed deficit. Can the authors elaborate more?

*We thank the reviewer for the comment. The general idea is that the deficit has an impact on the vertical shear, which in turn is a key parameter for turbulence generation and hence the deficit diffusion. We have added a respective comment after eq. 9. This comment is related to previous comment about the equations whether turbulence was taken into account or not.*

**Minor comments**

The following are minor comments to be considered by the authors if they please.

- Fig 1a, b: Consider using a discrete colour bar instead, as it may enhance readability.
  We thank the reviewer for the comment. To improve readability, we have instead ensured that the color bar is clearly labelled with its title next to it (as also suggested by Reviewer 1). Please see comment of Reviewer N1.

- The introduction section frequently refers to things that are yet to be introduced later in the manuscript. This may reduce the clarity for readers unfamiliar with the material.
  *We thank the reviewer for the comment. We revised the Introduction to reduce forward references and provide brief explanations at their first mention in order to improve overall clarity.*
  *"The commonly used wind turbine parameterisation (Fitch parametrisations (Fitch et al., 2012) is used .....the thust coefficient curve (CT -curve)..."*

- Lines 158-172: Can this be summarized? This part is very detailed, and not all these details are relevant to the main purpose of the manuscript.
  *We thank the reviewer for the comment. We revised the section describing the SAR data to enhance readability and have included additional methodological details as suggested by Reviewer 1 (please see response to comment of Reviewer 1).*

- Lines 178-182: Presenting this section as bullet points may improve readability.
  *We thank the reviewer for the comment. The section has been reformatted as a structured list of steps to enhance readability.*
  *"1. points with NRCS>1 are excluded,*
  *2. NRSC values in areas with water depth < 10 m are excluded ...*
  *4. Finally, the total ..."*

- In the caption of table A1, please briefly mention what is $\Delta T$ and $U10$. Are they spatial averaged values or are they taken at the FINO-1 site as in Fig. 2?

*We thank the reviewer for the comment. Indeed, Table A1 lists ΔT and U10 but the caption did not define them. We have now clarified in the caption that ΔT refers to the air–sea temperature difference (T2m – SST) and U10 refers to the 10 m wind speed.*
*"Table A1. Sentinel-1A and …. ΔT denotes to the air-sea temperature difference (T2m – SST) and U10 refers to the 10 m wind speed. The …."*

**Suggestions**

I have a few suggestions to the authors to consider if they please.

- The underlying physics assumptions of the presented model needs to be thoroughly revised.

*We thank the reviewer for the comment. We understand the reviewer's criticism and we have tried in the context of previous comments to explain that we had to find a compromise with respect to physical consistency to be able to solve the very technically demanding inversion problem. We could have avoided questions about physical consistency by using machine learning techniques, but we decided against it to keep stronger links with first order physical principles and to achieve better transparency.*

- A clear mention of the limitations of the proposed model would help the reader understand when it is suitable to use the model.
  *We thank the reviewer for the comment. We changed the title to make clear that this model was fitted to the situation in the German Bight. We also added a comment in the conclusion emphasizing this point. The title now reads*

  *"A semi-empirical model for near sea surface wind speed deficits downstream of offshore wind parks in the German Bight fitted to satellite synthetic aperture radar measurements"*

- Consider validating the model with SAR images for other sites than the North Sea.
  *We thank the reviewer for the comment. In the revised manuscript we added statements about limitations of the model and the fact that it was fitted for the situation in the German Bight. Apart from that, we do not have access to technical wind farm information outside of the North Sea.*

- The impact of turbine-induced turbulence should be accounted for.

*We thank the reviewer for the comment.*

*We tried to explain at different places in our response that turbulence is included implicitly in the parameterization of the model. We were playing with the idea to include a prognostic equation for TKE, but this would have created additional challenges with respect to the missing information about the vertical structure, e.g. concerning turbulence dissipation.*

*We added the following in the conclusions:*

*"We would like to emphasize that the tuning of the wake model to satellite data in the German Bight leads to limitations with regard to the general applicability. For example, the wind directions in the German Bight are predominantly from the north and west, which means that many situations are characterised by fully developed marine boundary layers and less by intermediate boundary layers associated with the close proximity of land. We think that a more thorough treatment of intermediate boundary layer cases would require an explicit prognostic inclusion of turbulent kinetic energy (TKE) in the model. Because of the substantially higher complexity and the corresponding challenges in the inversion we decided to address this issue in a separate study."*